# Effects of Exogenous Boron on Salt Stress Responses of Three Mangrove Species

**DOI:** 10.3390/plants14010079

**Published:** 2024-12-30

**Authors:** Jingjun Yang, Haihang Wei, Pifeng Lei, Jie Qin, Hongdeng Tian, Donghan Fan, Jihui Zhang, Zhenkai Qin, Xiaoying Huang, Xiu Liu

**Affiliations:** 1College of Life and Environmental Sciences, Central South University of Forestry and Technology, Changsha 410004, China; yangjj586@163.com; 2Guangxi Forestry Research Institute, Nanning 530002, China; whh190820@outlook.com (H.W.); qinjie82@163.com (J.Q.); thd198924@163.com (H.T.); 14787890454@163.com (J.Z.); 2009301018@st.gxu.edu.cn (Z.Q.); 3Qinzhou Forestry Research Institute, Qinzhou 535012, China; fandonghan@sina.com; 4Scientific Research Academy of Guangxi Environmental Protection, Nanning 530022, China; 15678185936@163.com

**Keywords:** boron, salt stress, mangrove, growth, morphology, physiology, anatomy

## Abstract

Salt stress is common but detrimental to plant growth, even in mangroves that live in saline areas. Boron (B) is an essential micronutrient that performs an important role in many functions in plants; however, its protective role under salt stress is poorly understood, especially in long-lived woody plants. In this study, we conducted an indoor experiment under simulated tidal conditions with four treatments (10‰ salinity, 40‰ salinity, 40‰ salinity + 100 μM B, and 40‰ salinity + 500 μM B) and three mangrove species (*Avicennia marina*, *Aegiceras corniculatum*, and *Bruguiera gymnorrhiza*) to investigate the effects of exogenous B on salt tolerance in plant growth, morphology, physiology, and leaf anatomy. The results showed that exogenous low-concentration B treatment (100 μM B) improved the performance of mangrove species under high salinity stress, especially in terms of physiology and leaf anatomy, while high-concentration B treatment (500 μM B) had adverse effects. Additionally, we found that the response to exogenous B varied among species in physiology and leaf anatomy, such as proline, malondialdehyde, activity of antioxidant enzymes, palisade tissue, and spongy tissue, which may be related to the salt tolerance of different species. This study may provide useful insights into the alleviation of salt stress by B in mangrove growth and development, which may facilitate mangrove cultivation and afforestation in a saline environment.

## 1. Introduction

Mangroves are mainly composed of evergreen shrubs or trees in the coastal intertidal zone in tropical and subtropical areas. The dense branches and tremendous roots of mangroves are the natural barriers to wind and waves [1,2], and they also have the function of providing food, biodiversity reserves, carbon sinks, and animal shelter [3,4]. However, due to climate change and anthropogenic perturbation, the area of mangroves has been greatly reduced, and the rate of destruction and disappearance of mangroves is much higher than that of tropical rainforests in terrestrial ecosystems [5]. Although many efforts have been made by researchers, organizations, and governments in the conservation of mangroves, the effect of afforestation and restoration to cope with the risk of mangrove disappearance and degradation have not met expectations. It is generally believed that mangroves have strong adaptability to salt stress, but salinity is still an important limiting factor affecting the recruitment and distribution of mangrove plants [6]. Reducing the effects of salt stress and improving the survival rate during cultivation, transplantation, and afforestation are important for the restoration of mangroves.

As typical halophytes, mangrove plants have specific adaptations to salinity in morphology, physiology, anatomy, and so on [6]. However, researchers believe that salt water is an ecological advantage for mangroves rather than a physiological need [7,8,9]. According to previous studies, the most suitable salinity for mangrove plant growth is 10–20‰, while the average salinity of seawater is 35‰. When the salinity of the growing environment exceeds an appropriate value, the growth of mangroves is inhibited or even dead [10]. For example, the growth of *Avicennia marina* and *Aegiceras corniculatum* decreased with an increase in salinity from 5‰ to 35‰ [11]. Plant height, biomass, and leaf area were all reduced under salt stress or even died under salt stress [12,13]. The physiological effects of salt stress on plants are mainly to affect osmotic pressure, disrupt the homeostasis of ions, produce reactive oxygen species (ROS), cause oxidative stress, and damage cell membranes [14,15]. Specifically, Na^+^ and Cl^−^ in the root, stem, and leaf tissues were significantly increased under salt stress, while the concentrations of Ca^2+^ and Mg^2+^ were decreased, which led to a decrease in chlorophyll (Chl) content and affected photosynthesis [16]. In addition, salt stress can increase antioxidant activity, such as superoxide dismutase (SOD) and catalase (CAT) [17], and the levels of malondialdehyde (MDA), indicating that salt stress causes oxidative stress to mangroves and affects membrane function [18].

Specialization of leaf anatomical features in mangroves plays an important role in salt regulation. The most consistent features of the mesophyll of mangroves are colorless water storage tissue, sclereids, and short tracheids terminating vein [19], and leaf of mangroves is dorsiventral that shows distinct palisade and spongy layers [20]. However, leaf anatomical features are also affected by salinity. Lin [21] found that with the increase of salinity, the thickness of palisade tissue and void in spongy tissue increased, while the number of stomata per unit area of leaves decreased in *Ae. corniculatum*. Under salt stress, both the stomatal area and cellular void were reduced in *Bruguiera. parviflora* [16]. Changes in leaf anatomy affect photosynthetic efficiency, which in turn affects the growth, morphology, and physiology of mangrove plants [22]. The response of leaf anatomy may show differences in salt stress in different mangrove species; however, we have a poor understanding of this, which may lead to low survival in cultivation, transplantation, and afforestation.

Boron (B) is one of the necessary microelements for plant growth, and can be combined with different molecules carrying cis-diol groups to form strong complexes, forming suitable spatial configurations and participating in plant metabolism [23,24]. Previous studies have shown that borates form ester bonds between the galactose residues of rhamnose-galactose II (RG II) monomers, thereby contributing to cell wall structure formation and functional maintenance [25,26]. In addition, B can also maintain the structural integrity of the plasma membrane through its association with membrane components [27], improve the nitrogen fixation function of plants, and contribute to the clearance of ROS [28]. Many studies have shown that addition of exogenous B can alleviate the symptoms of salt stress and improve the survival rate and growth status of plants under salt stress [27,29,30,31]. However, the response of long-lived woody plants (such as mangrove plants) to B supplementation under salt stress is not well understood, and the mechanism by which B alleviates the symptoms of salt stress in plants remains unclear. Here, we conducted an indoor experiment with four treatments (10‰ salinity, 40‰ salinity, 40‰ salinity + 100 μM B, and 40‰ salinity + 500 μM B) for three mangrove species. The aim of the present study was to investigate the growth, morphology, physiology, and leaf anatomy of salt-stressed mangroves and the regulatory function of B in improving mangrove performance under salt stress.

## 2. Results

### 2.1. Effects of B on Plant Growth

The addition of exogenous B exhibited a positive effect on height increment for all the mangrove species investigated in the three high salinity treatments (Figure 1a), although significant differences were only detected in the height increment of *B. gymnorrhiza* between T3 (12.16 ± 1.42 cm) and the treatment without exogenous B (T2, *p* < 0.05). Diameter increments of all treatments in all species were not significantly different (*p* > 0.05), although low concentration (100 μM) of B addition showed a higher value of the diameter growth (Figure 1b). There were also significant differences in leaf and root biomass among all treatments for all species (Figure 1c,d), except for the leaf biomass of *Av*. *marina* in T4 compared to T1 (*p* < 0.05). A low concentration of B addition had positive effects on leaf and root biomass, but a high concentration (500 μM) of B addition seemingly aggravated the effect of salt stress on plants.

### 2.2. Effects of B on Morphological Traits

There were no significant differences in the leaf area of different species among the different treatments (Figure 2a). Compared to T2 (salt stress without adding B), low concentration of B addition had a positive effect on leaf area in the three mangrove species, while high concentration of B addition had a negative effect on leaf area. When plants were subjected to salt stress, leaf thickness increased significantly in *Av. marina*, while *Ae. corniculatum* and *B. gymnorrhiza* showed the opposite pattern (Figure 2b). B addition significantly increased leaf thickness under salt stress in *B. gymnorrhiza* (*p* < 0.05), but *Av. marina* and *Ae. corniculatum* showed different responses to different concentrations of B addition. B addition had a positive effect on root diameter and total root area under salt stress, regardless of low or high concentration. Compared with T2, B addition increased the root diameter and total root area of these species by 4.3~61.7% (Figure 2c) and 2.4~30.7% (Figure 2d), respectively.

### 2.3. Effects of B on Plant Physiology

The total chlorophyll (Chl) content decreased significantly in the three species under salt stress compared to the unstressed controls (Figure 3a). The addition of low concentration of B to salt-treated seedlings significantly increased the Chl content (*p* < 0.05), but high concentration of B decreased it. Both MDA and Pro were increased in all species under salt stress, while B addition (both low and high concentration) decreased MDA as well as increased proline (Pro) (Figure 3b,c). Interestingly, *Av. marina* had the lowest content of MDA and Pro among the three species. The activities of the antioxidant enzymes SOD, CAT, and ascorbate peroxidase (APX) were significantly affected by salt stress (*p* < 0.05), and they increased significantly in the B addition treatments, except for SOD in T3 and APX in T4 of *Av. marina* (Figure 3d–f). Salt stress increased the content of Na^+^ and decreased the content of K^+^ in the three species (Figure 3g,h), which indicated that B addition significantly reduced the effect of salt stress (*p* < 0.05), and the higher the concentration, the lower the effect of salt stress.

### 2.4. Effects of B on Leaf Anatomy

In general, salt stress reduced leaf anatomical traits in the three species (Figure 4 and Table 1), especially in the upper epidermis (UEP), palisade tissue (PT), and spongy tissue (ST) in *Ae. corniculatum,* and *B. gymnorrhiza* (*p* < 0.05). The addition of B could improve leaf anatomical traits under salt stress, but it was not always effective. For example, the PT of *B. gymnorrhiza* significantly decreased in the B addition treatment (*p* < 0.05), regardless of concentration. However, high-concentration B addition had side effects on the UEP, hypodermis (HY), PT, and ST. Interestingly, salt stress and B addition had a significant influence on the leaf anatomical structure in *Ae. corniculatum,* and *B. gymnorrhiza*, but not in *Av. marina* (*p* > 0.05).

## 3. Discussion

When plants suffer salt stress, their growth, morphology, physiology, etc., are affected, even in mangroves that grow in saline environments. However, the effects of salinity on mangrove plants are often ignored, especially during cultivation and afforestation. There are a lot of studies on alleviating the effects of salt stress by adding exogenous substances [27,29,30,31], but to our knowledge, there are no studies on mangroves. In this study, by adding exogenous B under salt stress conditions in an indoor experiment, we analyzed whether the addition of exogenous B helped mangrove plants alleviate salt stress in four aspects: growth, morphology, physiology, and anatomy.

Our results showed that the addition of low-concentration exogenous B (T3, 100 μM) had positive effects on the height, diameter, leaf, and root of these plants under salt stress conditions, but not under high-concentration B treatment (T4, 500 μM). These results were similar to those of a previous study, which revealed that B addition improved plant growth, but not significantly compared with the absence of the addition [31]. Several other studies have also demonstrated a B-induced improvement in the growth parameters of plants growing under saline conditions [30,32]. The content of B can affect the intensity of photosynthesis and the transport of photosynthetic products, thus affecting the growth of individual plants [33,34,35,36]. Additionally, the responses of the growth performance of plants to stress are much slower than the physiological level, which may explain why most of the growth parameters were not significant. The increments in growth were only statistically significant in *B. gymnorrhiza* with low exogenous B compared with other treatments. This may be because different tree species have different sensitivity to different concentrations of B. As a trace element, the positive effect of B on plants exists only within a small range, and different species have different requirements for B [26,37]. Moreover, excessive B will hinder the flow of phloem [23], which can aggravate the effect of salt stress on plants.

Salt stress also reduces the morphological trait values, like leaf area, leaf thickness, root diameter, and total root area. The improved salt stress effect of low concentration of B and the aggravated salt stress effect of high concentration of B were also observed in leaf traits. However, B addition had a positive effect on root diameter and total root area under salt stress, regardless of low or high concentration. In the salt stress experiment of rose, the leaf area and relative water content increased with an increase in the concentration of B [30]. This result might indicate that the sensitivity specificity of B concentration exists not only in different species but also in different parts of the same plant. Studies have shown that B is crucial for root growth and development [38] because it is conducive to the expansion and division of cells in the meristem [23]. This may result in the root being less sensitive to B concentration than the leaves.

Analysis of plant physiology showed that the content of Chl decreased significantly in the three species under salt stress compared with the unstressed controls. While the addition of low-concentration B to salt-treated seedlings significantly increased Chl content (*p* < 0.05), high-concentration B decreased it. Both MDA and Pro were increased in all species under salt stress, while B addition (both low and high concentration) decreased the MDA and increased the Pro. Interestingly, *Av. marina* had the lowest content of MDA and Pro among the three species. This ability to maintain a low MDA content in a high salt environment may help *Av. marina* to have high salt tolerance. Activities of the antioxidant enzymes SOD, CAT, and APX were significantly affected by salt stress (*p* < 0.05), and they increased significantly in the B addition treatments. In a previous study, the addition of exogenous B had positive effects on the growth, biomass, and Chl content of soybean (*Glycine max* L.), but there was no significant difference compared to the control. At the same time, the contents of MDA and H_2_O_2_ were decreased by B addition, and the contents of ascorbic acid and glutathione were increased [31]. This indicates that B addition may contribute to plants removing ROS and increasing the content of REDOX substances, which alleviates the oxidative stress caused by salt stress [27]. Salt stress increased the content of Na^+^ and decreased the content of K^+^ in the three species. B addition significantly reduced the content of Na^+^ and raised the content of K^+^ in our study. Bonilla et al. [29] studied the effects of B addition on the growth of pea (*Pisum sativum*) seedlings under different concentrations of salt stress (0–150 mM NaCl) and found that the addition of B could counteract the inhibitory effect of salt stress. This is helpful for the recovery of Na^+^, K^+,^ and Fe^2+^ in stems and roots, and reduces the concentration of Na^+^. K^+^ is an important ion for maintaining cell penetration, and a high concentration of Na^+^ can preempt the binding sites of K^+^ and Ca^2+^, making the cell osmotic pressure unbalanced, making it difficult to absorb water from the outside world, and ultimately inhibiting the growth and development of individuals [15]. Therefore, the addition of exogenous B can help plants relieve salt stress at the physiological level.

The addition of B could influence some leaf anatomical structures under salt stress among the three mangrove species. Specifically, the addition of a low concentration of B significantly improved the PT, but not the ST of *Ae. corniculatum*. On the contrary, it improved the ST but not the PT of *B. gymnorrhiza*. However, the addition of high concentrations of B showed a less positive or even negative effect on PT and ST than low concentrations of B. A study on African mahogany showed that excessive B levels could lead to disorganized palisade parenchyma and a reduction in intracellular spaces [39], which is similar to our results. Interestingly, salt stress and excessive B had a significant influence on the leaf anatomical structure in *Ae. corniculatum,* and *B. gymnorrhiza*, but not in *Av. marina*. In the natural *Av. marina* population, there was no correlation between the quantitative characteristics of leaf tissue and salinity, but this relationship existed in *Kandelia obovata*, which belongs to the same family (Rhizophoraceae) as *B. gymnorrhiza* [40]. A possible reason for this is that *Av. marina* has a higher salt tolerance than *Ae. corniculatum,* and *B. gymnorrhiza*. Therefore, its leaf anatomy is also less sensitive to salt stress or/and B than the other two species. Future studies should incorporate more species or methods (e.g., transcriptomics and metabonomics) to reveal the mechanism of the effect of exogenous B on mangrove growth under salt stress, which may provide a scientific reference for the cultivation and afforestation of mangroves.

## 4. Materials and Methods

### 4.1. Experimental Design

The experiment was conducted in a cultivated greenhouse at the Guangxi Forestry Research Institute (108°21′24″ E, 22°55′1″ N). Three mangrove species, consisting of two cryptic viviparous species [*Avicennia marina* (Forsk.) Vierh.] and [*Aegiceras corniculatum* (L.) Blanco.] and one true viviparous species [*Bruguiera gymnorrhiza* (L.) Savigny.] were selected because they are commonly found in mangrove forests in this region. One year old saplings were bought from the same nursery company and had similar heights and diameters for each species. All saplings were placed in an indoor tidal simulation system (Figure 5a,b) during the experiment. The indoor tidal simulation system can simulate the flood tide and ebb tide through the timing device to create an environment that is similar to the mangrove natural habitat (Figure 5c,d). The flooding time was set to 8h during the day or night, and it was rotated once a week. A total of four treatments were set for each tree species in this experiment: salinity 10‰ (T1), salinity 40‰ (T2), salinity 40‰ + 100 μM B (T3), and salinity 40‰ + 500 μM B (T4). Each tidal simulation system had one treatment containing 10 saplings; thus, there were 120 saplings in total. The form of exogenous B was a boric acid solution (Sigma Aldrich (Shanghai) Trading Co., Ltd., Shanghai, China). Exogenous B was added once a week during the ebb tide period in this experiment, and 20 mL was added each time.

### 4.2. Measurements

#### 4.2.1. Growth Parameters

The sapling height and diameter were measured at the beginning (18 June 2023) and the end (18 December 2023) of this experiment, respectively. Sapling height of *Av. marina* and *Ae. corniculatum* was measured from ground to apical meristem by using linear tape, and the stem diameter on the base was measured 5 cm above ground with a vernier caliper (0.01 mm) [41]. *B. gymnorrhiza* is a true viviparous mangrove; hence, the seedling basal diameter was measured from the stem at the upper end of the hypocotyl. After the experiment was finished, three plants (*n* = 3) were randomly selected to determine biomass and morphological traits. The attached soil was carefully washed and the plants were separated into roots and leaves. The leaves and roots were dried after the morphological traits were determined (see Section 4.2.2). These plant parts were placed into envelopes and dried in a 105 °C oven for 30 min, followed by further drying at 80 °C until a constant weight was achieved [4]. After cooling in an oven to room temperature, the dry biomass of the plant parts was weighed using an electronic balance (0.01 g).

#### 4.2.2. Morphological Traits

Leaf and root samples were placed in a transparent tray on a scanner (image resolution: 400 dpi). The morphological traits of leaves and roots, including leaf area, root diameter, and total root area, were analyzed using WinRHIZO 2013 software (Regent Instruments Inc., Québec City, QC, Canada) using images obtained [42]. Leaf thickness was determined by measuring leaf anatomy (see Section 4.2.4). All the determination of morphological traits was completed within one day.

#### 4.2.3. Physiological Parameters

For each treatment, three saplings (*n* = 3) were randomly selected for the determination of physiological parameters, and each sapling was collected from 5 to 6 leaves.

Total Chl was determined spectrophotometrically using 80% acetone as the solvent [43].

The Pro content was estimated using ninhydrin under acidic conditions [44], and the standard curve was established using a high-purity standard product (Shanghai Yuanye Bio-Technology Co., Ltd., Shanghai, China, HPLC ≥ 99%). The MDA level was determined by the Thiobarbituric Acid Reactive Substances Assay [45]. For more details, please refer to Alharby et al. [31].

The activity of three antioxidant enzymes was determined, including SOD (EC: 1.15.1.1), CAT (EC: 1.11.1.6), and APX (EC: 1.11.1.11). The SOD activity assay was performed according to the method described by Giannopotitis and Ries [46]. CAT activity was determined based on the method described by Cakmak and Marschner [47]. APX activity was determined according to Nakano and Asada [48]. For further details, refer to Tavallali et al. [27].

The dry leaf was ground to a fine powder, and 50 mg was weighed and placed into a test tube to measure the ion content. Ultra-pure water (10 mL) was added to each tube and heated to 100 °C for 4 h in a water bath. After the tubes were cooled, the resulting extracts were filtered, diluted, and used to determine Na^+^ and K^+^ content on a flame photometer (AP1500, Aopu, Guangzhou, China) [49].

#### 4.2.4. Leaf Anatomy

Determination of leaf anatomical structure was performed as described by Li [41]. In brief, complete and mature leaves were collected, and then a scalpel was used to take the transverse tissue of the middle of the leaves across the main vein (~10 mm). All samples were fixed in 20 times the volume of Formalin-Aceto-Alcohol fixative (70%) (Sigma Aldrich (Shanghai) Trading Co., Ltd., Shanghai, China). The samples were stained with Safranin and Fast Green after paraffin embedding. The images of the paraffin section were scanned by microscope (DP26, OLYMPUS, Tokyo, Japan), and SildeViewer "http://www.3dhistech.com (accessed on 16 December 2024)” was used to observe and measure leaf anatomy with the image. For each image, three visual fields were randomly selected for observation and measurement.

### 4.3. Data Analysis

To compare the effects of exogenous B on growth, morphology, physiology, and anatomy in each mangrove species, all the indexes in different treatments were analyzed by means of one-way analysis of variance (ANOVA) with Tukey HSD to test whether there were significant differences among different treatments in each species (*p* < 0.05). Statistical analyses were performed using the software R project (R 4.4.0) [50].

## Figures and Tables

**Figure 1 plants-14-00079-f001:**
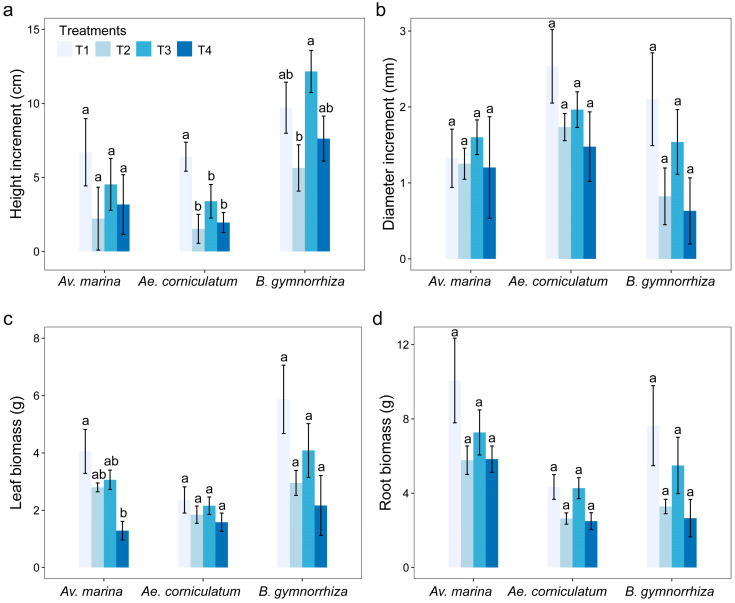
Increments of height (**a**), increments of diameter (**b**), leaf biomass (**c**), and root biomass (**d**) in different treatments of the three mangrove plants. Bars represent means ± SE. T1, salinity 10‰; T2, salinity 40‰; T3, salinity 40‰ + 100 μM B; T4, salinity 40‰ + 500 μM B. Different letters indicate significant differences among different treatments at the same species, *p* < 0.05 (Tukey HSD test).

**Figure 2 plants-14-00079-f002:**
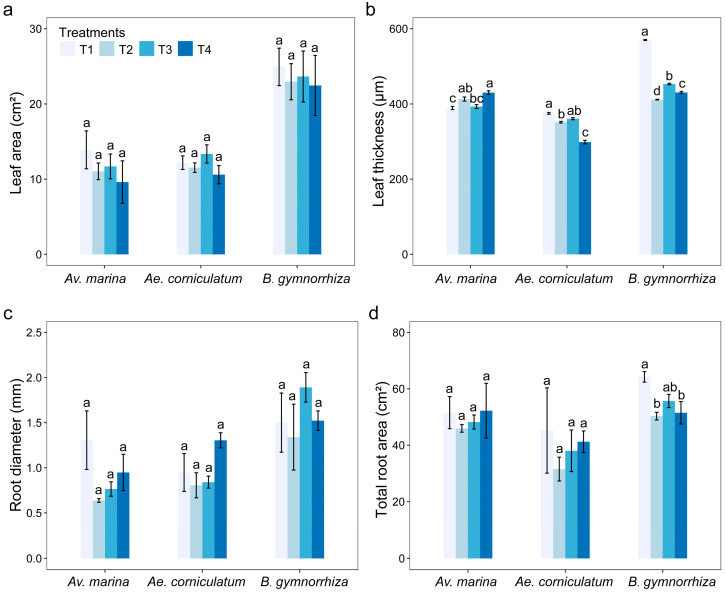
Leaf area (**a**), leaf thickness (**b**), root diameter (**c**), and total root area (**d**) in different treatments of the three mangrove plants. Bars represent means ± SE. T1, salinity 10‰; T2, salinity 40‰; T3, salinity 40‰ + 100 μM B; T4, salinity 40‰ + 500 μM B. Different letters indicate significant differences among different treatments at the same species, *p* < 0.05 (Tukey HSD test).

**Figure 3 plants-14-00079-f003:**
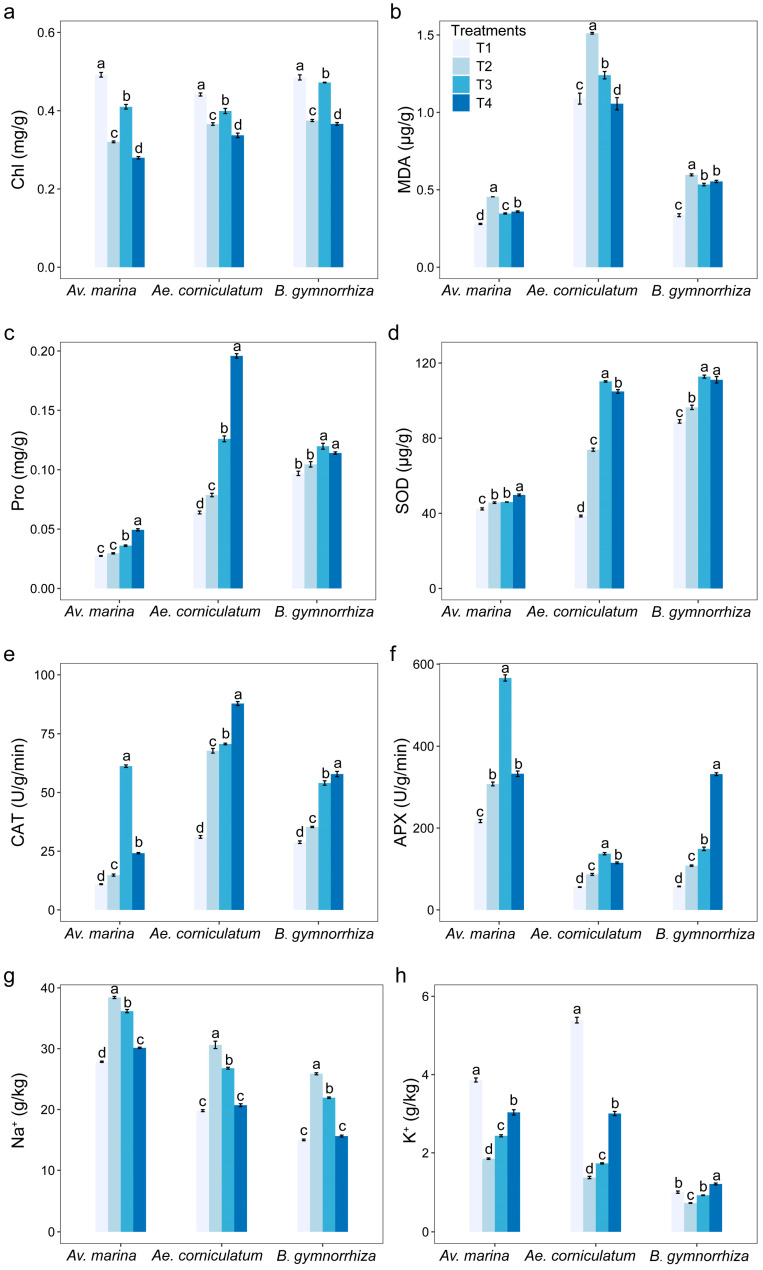
The content of Chl (**a**), MDA (**b**), and Pro (**c**); activities of SOD (**d**), CAT (**e**), and APX (**f**); Na^+^ (**g**) and K^+^ content (**h**) in different treatments of the three mangrove plants. Bars represent means ± SE. T1, salinity 10‰; T2, salinity 40‰; T3, salinity 40‰ + 100 μM B; T4, salinity 40‰ + 500 μM B. Different letters indicate significant differences among different treatments at the same species, *p* < 0.05 (Tukey HSD test).

**Figure 4 plants-14-00079-f004:**
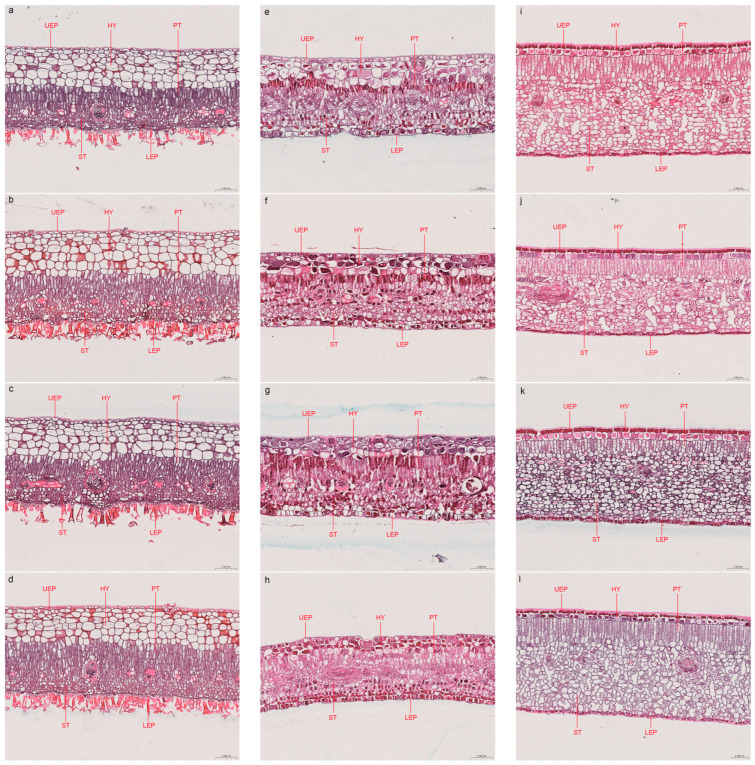
Photomicrographs of leaf anatomy of *Avicennia marina* (**a***–***d**), *Aegiceras corniculatum* (**e***–***h**), and *Bruguiera gymnorrhiza* (**i***–***l**) in different treatments (T1 to T4, respectively). UEP, upper epidermis; HY, hypodermis; PT, palisade tissue; ST, spongy tissue; LEP, lower epidermis. T1, salinity 10‰; T2, salinity 40‰; T3, salinity 40‰ + 100 μM B; T4, salinity 40‰ + 500 μM B.

**Figure 5 plants-14-00079-f005:**
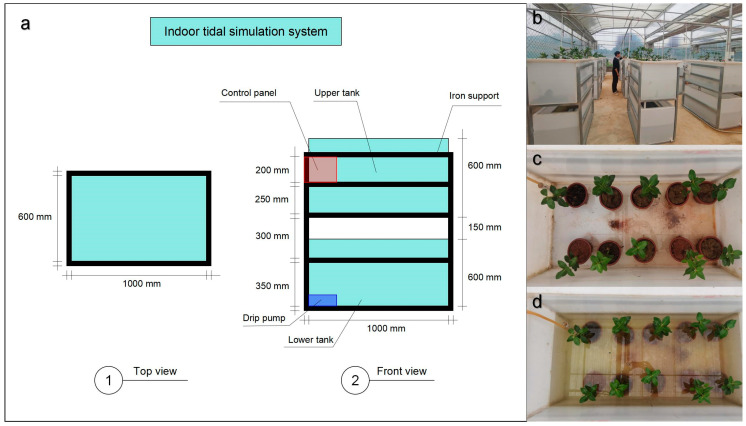
Illustration of the indoor tidal simulation system (**a**), actual picture of indoor tidal simulation system (**b**), scenario of ebb tide (**c**), and scenario of flood tide (**d**).

**Table 1 plants-14-00079-t001:** The leaf anatomical traits of three mangrove plants in different treatments.

Index	Treatments	*Avicennia marina*	*Aegiceras corniculatum*	*Bruguiera gymnorrhiza*
UEP (μm)	T1	13.37 ± 0.94 a	17.63 ± 0.56 a	14.53 ± 0.64 bc
	T2	11.80 ± 0.35 a	14.70 ± 0.25 b	13.73 ± 0.19 c
	T3	14.53 ± 0.83 a	17.60 ± 0.15 a	18.93 ± 0.81 a
	T4	12.03 ± 0.50 a	16.70 ± 1.01 ab	17.10 ± 0.44 ab
HY (μm)	T1	176.80 ± 1.36 a	128.47 ± 2.61 a	35.90 ± 1.34 a
	T2	187.77 ± 9.19 a	79.13 ± 1.27 b	33.77 ± 0.73 a
	T3	171.67 ± 13.37 a	79.33 ± 4.23 b	35.23 ± 1.41 a
	T4	169.50 ± 7.41 a	66.90 ± 2.64 b	33.20 ± 0.53 a
PT (μm)	T1	140.53 ± 9.40 a	98.90 ± 3.82 b	115.67 ± 2.81 a
	T2	128.53 ± 1.73 a	77.60 ± 2.40 c	87.77 ± 1.01 b
	T3	140.47 ± 6.79 a	133.37 ± 1.58 a	75.83 ± 0.95 c
	T4	151.47 ± 2.47 a	73.23 ± 1.84 c	61.27 ± 1.45d
ST (μm)	T1	48.70 ± 3.67 a	101.07 ± 4.35 a	378.57 ± 2.98 a
	T2	47.20 ± 2.78 a	78.00 ± 3.73 b	246.27 ± 1.52 d
	T3	54.53 ± 1.66 a	88.13 ± 3.27 ab	302.20 ± 5.34 b
	T4	58.87 ± 2.74 a	72.53 ± 3.76 b	283.17 ± 1.05 c
LEP (μm)	T1	9.23 ± 0.69 a	21.63 ± 0.98 a	9.63 ± 0.66 b
	T2	11.33 ± 1.33 a	17.57 ± 1.37 a	10.23 ± 0.46 b
	T3	9.30 ± 0.75 a	21.83 ± 1.86 a	14.00 ± 1.21 a
	T4	8.80 ± 0.32 a	19.47 ± 0.12 a	10.17 ± 0.67 b
CTR (%)	T1	36.16 ± 2.74 a	26.37 ± 0.85 b	20.29 ± 0.47 a
	T2	31.13 ± 0.32 a	22.08 ± 0.57 c	21.32 ± 0.29 a
	T3	35.77 ± 2.02 a	36.99 ± 0.71 a	16.74 ± 0.26 b
	T4	35.23 ± 0.89 a	24.56 ± 0.99 b c	14.23 ± 0.26 c
SR (%)	T1	12.50 ± 0.84 a	26.96 ± 1.16 a	66.41 ± 0.41 a
	T2	11.43 ± 0.66 a	22.21 ± 1.19 a	59.85 ± 0.26 b
	T3	13.86 ± 0.26 a	24.44 ± 0.92 a	66.72 ± 1.11 a
	T4	13.68 ± 0.51 a	24.28 ± 1.03 a	65.78 ± 0.21 a

The data are presented as mean ± SE. Different letters indicate significant differences among different treatments in the same species, *p* < 0.05 (Tukey HSD test). UEP, upper epidermis; HY, hypodermis; PT, palisade tissue; ST, spongy tissue; LEP, lower epidermis; CTR, cell tension ratio; SR, spongy tissue ratio. T1, salinity 10‰; T2, salinity 40‰; T3, salinity 40‰ + 100 μM B; T4, salinity 40‰ + 500 μM B.

## Data Availability

The data will be made available by the authors upon request.

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
