# Peer review of "Effects of Exogenous Boron on Salt Stress Responses of Three Mangrove Species"

_plants, 2024, doi:10.3390/plants14010079_

Round 1

Reviewer 1 Report

Comments and Suggestions for Authors

The authors describe very interesting relationships. However, the manuscript does not read well, in my opinion it needs refinement. I have included all my observations, recommendations or comments in the attached file.

Reviewer 2 Report

Comments and Suggestions for Authors

In this manuscript, the authors reported their investigation on the effects of two levels of B applications (100 μM & 500 μM) on the growth and physiological parameters of three mangrove species, namely Avicennia marina, Aegiceras corniculatum, and Bruguiera gymnorrhiza, subjected to salt stress (40% salinity). While some interesting findings have been presented, careful recheck of the manuscript is required - considering some statements seem confusing/vague, and there are a few important errors as described below. Novelty of findings should be more explicitly highlighted too. Below are my comments:

1.     TITLE:  I would suggest that it be revised to something along the lines of “Effects of exogenous boron on salt stress responses of three mangrove species” or something like that. The original title seems weak/inaccurate because looking at the growth data in Figure 1, the low B treatment (100 μM) only showed positive effect in one case, i.e., by increasing the height of B. gymnorrhiza. All other comparisons between T2 vs T3 for all four parameters in Figure 1 lack statistical significance, thus not clearly supporting the claim that “Low Concentration of Exogenous Boron Enhances Salt Tolerance in Three Mangrove Species”.

2.     Important: Figure 4 caption is not matching what is shown in Figure 4. In fact, the current caption for Figure 4 is identical to the caption of Figure 3. Please rectify this.

3.     ABSTRACT:

·       Instead of mentioning “low concentrations of exogenous B” (line 22) and “high B concentrations” (line 24), it would be clearer to specify directly the B concentrations instead.

·       Line 28: It is unclear what “four dimensions” refers to. It would be good to replace it with a more accurate word.

4.     INTRODUCTION:

·       Instead of mentioning “the lipid membrane oxide malondialdehyde (MDA),” (line 62), it would be clearer to just mention “levels of malondialdehyde (MDA)” instead. The original expression is ambiguous/inaccurate, considering that MDA is a byproduct of lipid peroxidation occurring in the membrane. The original expression may imply that MDA is a preexisting or naturally occurring oxide in the membrane.

·       ROS” has been introduced with its full term twice. Please see lines 57 and 85. Furthermore, ROS often refers to “reactive oxygen species”. But the full term indicated by the authors is “reactive oxides”. Please recheck whether it is correct.

5.     RESULTS:

·       Lines 95-106 (first paragraph): A major issue here is that some statements seem inaccurate, considering the lack of statistical significance in the data shown in Figure 1.

·       For example, “Height increments of Av. marina … and Ae. corniculatum … in T1 was bigger than other treatments…” (lines 95-97) – While it is true for Ae. corniculatum, it is untrue for Av. marina. For Av. marina, there is no statistically significant difference among T1, T2, T3 and T4. So, for Av. marina, it is not valid to claim that height increment in T1 is the biggest.

·       Lines 97-99: “The addition of exogenous B helped … only the increment of B. gymnorrhiza was significantly bigger than the treatment without exogenous B (T2, p < 0.05)” – This statement is also only partly valid and should be revised to make it more accurate. Based on Figure 1a, the statement is only true for 100 μM; but it is untrue for 500 μM, as there is no statistically significant difference between T2 and T4.

·       Line 101: “low concentration (100 μM) of B addition facilitated the diameter growth” – This statement is not valid due to lack of statistically significant differences between T2 and T3 in all three species as shown in Figure 1b.

·       Lines 102-106: The two statements here also need to be rechecked and rectified, considering whether there is any statistical significance in the data. Importantly, for this statement “Salt stress reduced the leaf and root biomass … but that had no significant differences …” – If there are no statistically significant differences, please try to avoid using the word “reduced” here as it is somewhat misleading/self-contradictory.

·       Lines 114-116: “Compared to T2 … low concentration … of B addition increased the leaf area … high concentration … of B addition decreased the leaf area.” This statement is inaccurate due to lack of statistically significant differences; it should be revised.

·       Lines 120-121: “Boron addition mitigated the reduction of salt stress in root diameter and total root area, regardless of low or high concentration” – Try to be consistent and don’t randomly switch between “B” and “boron”. Importantly, this statement is again inaccurate due to lack of statistically significant differences; it should be revised.

·       Lines 136-138: “Activities of the antioxidant enzymes SOD, CAT, and APX were significantly affected by salt stress (p < 0.05), and they increased significantly in B addition treatments (Figs. 4d-f).” – The statement is inaccurate. For SOD, Av. marina T3 is not significantly different compared with Av. marina T2. For APX, Av. marina T4 is not significantly different compared with Av. marina T2. So, the statement needs to be revised to more accurately reflect the data shown.

·       Lines 139-140: “Born addition significantly reduced the effect of salt stress (p < 0.05), and the higher the concentration, the less effect of salt stress.” – It is unclear what “effect” refers to here. Ideally, please indicate which chart the statement refers to. Due to lack of clarity, it is difficult to check whether P is really < 0.05.  Born” should be “B” or “Boron”. Please revise.

·       Lines 148-154:

o   It is preferable to revise the text here, taking into consideration which data are statistically significant, which not.

o   Abbreviations “UEP, HY, PT, ST, LEP, CTR and SR” should be introduced upon first mention too.

o   Fig. 5” (line 149) should be “Fig. 4” instead.

o   P<0.05” (line 151) – “P” is sometimes shown as a lowercase “p” in the text, sometimes uppercase; please standardize.

o   Line 152: “… B addition had the side effect in UEP, HY, PT, ST.” – Please revise the sentence so that it is clear what “side effect” refers to.

·       Line 160: “Table 1. Summary of the leaf anatomic structure…” can be revised to “Table 1. The leaf anatomic traits or parameters…”. Also, table title is too long. It would be easier to read if some of the relevant info can be moved to the footnote of the table.

6.     DISCUSSION:

·       Lines 169-170: “For mangroves, there are few studies on alleviating the effects of salt stress by adding exogenous substances.” – Statement is vague. Please be more specific about what “exogenous substances” are those. If relevant, please also indicate which mangrove species are those. Cite the references of those studies too.

·       Lines 174-176: Please revise “Our results showed that addition of low concentration exogenous B (T3, 100 μM) helped…but not in the high concentration B treatment (T4, 500 μM).” Please take into consideration whether the data show any statistically significant differences.

·       Lines 195-196: “…B addition mitigated the reduction of salt stress in root diameter and total root area, regardless of low or high concentration.” – Please recheck. This statement is inaccurate because of the lack of statistically significant differences among T2-T4, as shown in the figures.

·       Lines 212-214: “… exogenous B promoted the growth, biomass and chlorophyll content of soybean … no significant difference compared with the control.” – This sentence is confusing – if there is no statistically significant difference, it should not be regarded to have “promoted” those parameters. Please recheck and revise it.

·       Lines 219-221: “Born addition significantly reduced the effect of salt stress (p<0.05), and the higher the concentration, the less effect of salt stress.” – This sentence appears in two locations in the text – the other location is on lines 139-140. In both locations, the sentence is 100% identical word by word. Please avoid “recycling” sentences in the text.

·       Line 222: “(0-150 Mmol NaCl)” - Please check whether it should be “Mmol” or “mmol”.

·       Lines 233-234: “… high concentrations B showed a side effects on PT and ST” – Sentence unclear. Please revise so that it is clear what “side effects” refers to.

7.     MATERIALS AND METHODS:

·       Lines 270-271: “(June 18th) … (December 18th)” - Please indicate whether the dates indicated refer to the year 2024.

·       Line 278: “(see 2.2.2).” should be “(see 4.2.2)”.

·       Line 286: “using the WinRHIZO 2013” should be “using the WinRHIZO 2013 software”.

·       Lines 287-288: “Leaf thickness (LT) was determined by leaf anatomy (see 4.2.4).” – This sentence sounds unclear. Please revise it.

8.     Throughout the manuscript (both text and figure/table captions), the symbol ‰ has been used to indicate percentage. Please rectify. For examples, please see lines 19, 51, 54, and 260-261.

9.     Some scientific names of the mangrove species are not in italics. Please rectify. For examples, please see lines 95, 233, 238, and 273.

10.  For the scientific names of mangrove species, the species names should begin with a lowercase letter. Please rectify. For examples, please see lines 117, 135, 208, 210, 232-233, and 272.

11.  All abbreviations should be introduced with their full terms on first mention. Please rectify. Please see lines 130 (“Chl”), 133 (“Pro”) and 312 (“FAA”). On the other hand, an abbreviation used only once may not need to be introduced at all, such as “TBARS” on line 297.

12.  Treat” appearing in Table 1 and Figures 1-3 should be changed to “Treatments”.

13.  Please recheck the manuscript carefully for typo errors, for examples:

·       slat” (should be “salt”) (line 55).

·       may showed” (should be “may show”) (line 74)

·       unstessed” (should be “unstressed”) (line 102)

·       speices” (should be “species”) (line 103)

·       thichness” (should be “thickness”) (line 118)

·       born” (should be “boron” or “B”) (lines 139, 219, and 261).

14.  The authors should explain briefly somewhere in the manuscript the rationales for choosing 40% for salinity stress, as well as for choosing 100 μM and 500 μM B instead of any other levels of B concentrations.
